# Effects of early life adversity on immediate early gene expression: Systematic review and 3-level meta-analysis of rodent studies

Heike Schuler[1,2☯], Valeria Bonapersona[1☯]*, Marian Joëls[1,3‡], R. Angela Sarabdjitsingh[1‡]

**1** Department of Translational Neuroscience, University Medical Center Utrecht Brain Center, Utrecht University, Utrecht, The Netherlands, **2** Integrated Program in Neuroscience, McGill University, Montréal, QC, Canada, **3** University Medical Center Groningen, University of Groningen, Groningen, The Netherlands

☯ These authors contributed equally to this work.
‡ MJ and RAS also contributed equally to this work.
* v.bonapersona-2@umcutrecht.nl

**Data Availability Statement:** All data files are available freely available at https://osf.io/qkyvd/.

**Funding:** MJ (Consortium of Individual Development, which is funded through the

## Abstract

Early-life adversity (ELA) causes long-lasting structural and functional changes to the brain, rendering affected individuals vulnerable to the development of psychopathologies later in life. Immediate-early genes (IEGs) provide a potential marker for the observed alterations, bridging the gap between activity-regulated transcription and long-lasting effects on brain structure and function. Several heterogeneous studies have used IEGs to identify differences in cellular activity after ELA; systematically investigating the literature is therefore crucial for comprehensive conclusions. Here, we performed a systematic review on 39 preclinical studies in rodents to study the effects of ELA (alteration of maternal care) on IEG expression. Females and IEGs other than cFos were investigated in only a handful of publications. We meta-analyzed publications investigating specifically cFos expression. ELA increased cFos expression after an acute stressor only if the animals (control and ELA) had experienced additional hits. At rest, ELA increased cFos expression irrespective of other life events, suggesting that ELA creates a phenotype similar to naïve, acutely stressed animals. We present a conceptual theoretical framework to interpret the unexpected results. Overall, ELA likely alters IEG expression across the brain, especially in interaction with other negative life events. The present review highlights current knowledge gaps and provides guidance to aid the design of future studies.

## Introduction

Synaptic connections in the brain are continuously altered, including via gene expression, to accommodate experiences, thereby preparing the organism to deal with future events [1–3]. This potential for adaptation, called neuronal or synaptic plasticity, is prominently present during critical periods early in life [4]. For this reason, adverse experiences throughout childhood–such as physical, sexual or emotional abuse–have far-reaching effects on an individual's brain function and structure, and consequently on cognition and behavior [5–7]. It is therefore

Gravitation program of the Dutch Ministry of Education, Culture, and Science and the Netherlands Organization for Scientific Research; grant number: 024.001.003). RAS (ZoNmW program MKMD; grant number 114024135). The funders had no role in study design, data collection and analysis, decision to publish, or preparation of the manuscript.

**Competing interests:** The authors have declared that no competing interests exist.

not surprising that *early-life adversity* (ELA) is consistently associated with an increased risk for psychopathologies later in life, including major depressive disorder (MDD), post-traumatic stress disorder (PTSD), and schizophrenia [8, 9].

To investigate the mechanisms underlying the effects of ELA on brain and behavior, several models of alteration of maternal care in rodents have been developed [10, 11]. These models consistently show that ELA leads to fundamental remodeling of stress-sensitive brain regions, which in turn may be linked to altered function [12, 13]. For example, ELA has been reported to modify the regulatory response of the *hypothalamic-pituitary-adrenal* (HPA) axis, an essential part of the organism's stress response system [14, 15]. Furthermore, rodents exposed to ELA display a robust behavioral phenotype characterized by enhanced anxiety-like behavior, changes in memory formation, and decreased social behavior [16–19]. Overall, this evidence highlights that ELA leads to structural, functional and behavioral alterations in the rodent brain, yet the events giving rise to the said alterations remain unclear.

Immediate-early genes (IEGs), such as *cFos* (alias *Fos*), *Egr1* (alias *Zif-268*, *NGFI-A*, *Krox-24*) and *Arc* (alias *Arg3.1*), provide a potential link between experience-induced cellular activity in the brain and the resulting long-term changes in neurons and synapses. IEGs are immediately and transiently expressed in response to extracellular calcium influx, as occurs when an action potential is fired [20]. Among the IEGs, *cFos* is most often studied; it forms the activator protein-1 (AP1) by dimerization with a *Jun*-family transcription factor [21]. The AP1 complex initiates the transcription of other late genes, which result in long-lasting changes of cellular physiology. Consequently, a strong relationship between IEG expression and neuronal activity is observed, with increases in neuronal activity being accompanied by increased IEG expression [20]. For decades, IEGs have been a prominent tool for mapping neuronal activity in rodents by means of immunohistochemistry (IHC) and in-situ hybridization (ISH) due to their brain-wide expression. More recently, IEGs have been increasingly investigated for their protein properties, in particular with respect to synaptic plasticity [22].

Whereas the downstream products of IEGs are diverse (*e.g.*, transcription factors, postsynaptic proteins, secretory factors), their functions are surprisingly homogeneous and can mostly be related to cellular processes, such as dendrite and spine development; synapse formation, strength and elimination; and regulation of the excitatory/inhibitory balance ([3]; Fig 1). In line with this functional similarity, knockouts (Kos) of several different IEGs affect behavior and synaptic plasticity in a similar manner. More specifically, system-wide *Arc*-KO and *Egr1*-KO, as well as central nervous system-specific *cFos*-KO mice all display behavioral impairments in learning and memory as well as deficits in long-term potentiation or depression, underscoring the necessity of IEGs for memory formation and retention [23–25]. In addition, many neuropsychiatric disorders characterized by memory impairments, such as major depressive disorder, post-traumatic stress disorder and schizophrenia, have also been shown to feature a dysregulation of activity-dependent transcription [26]. Interestingly, the risk to develop any of these disorders is increased by exposure to ELA, further indicating a potential causal interaction between ELA, IEGs and mental health [8, 9].

While numerous studies have used IEGs to identify differences in cellular activity after ELA, the study designs are heterogeneous, and findings are seemingly discrepant. Reviewing the available literature will provide a clearer picture of the effects of ELA on IEG expression and will aid future development of study designs by identifying sources of heterogeneity within and between experiments. To that end, we performed a systematic review to synthesize the available evidence and explore outcomes in a sex-, gene- and region-specific manner. A meta-analysis was then conducted on a subset of the data based on *a priori* determined thresholds. We hypothesized that ELA as alteration of maternal care leads to an exaggerated increase in

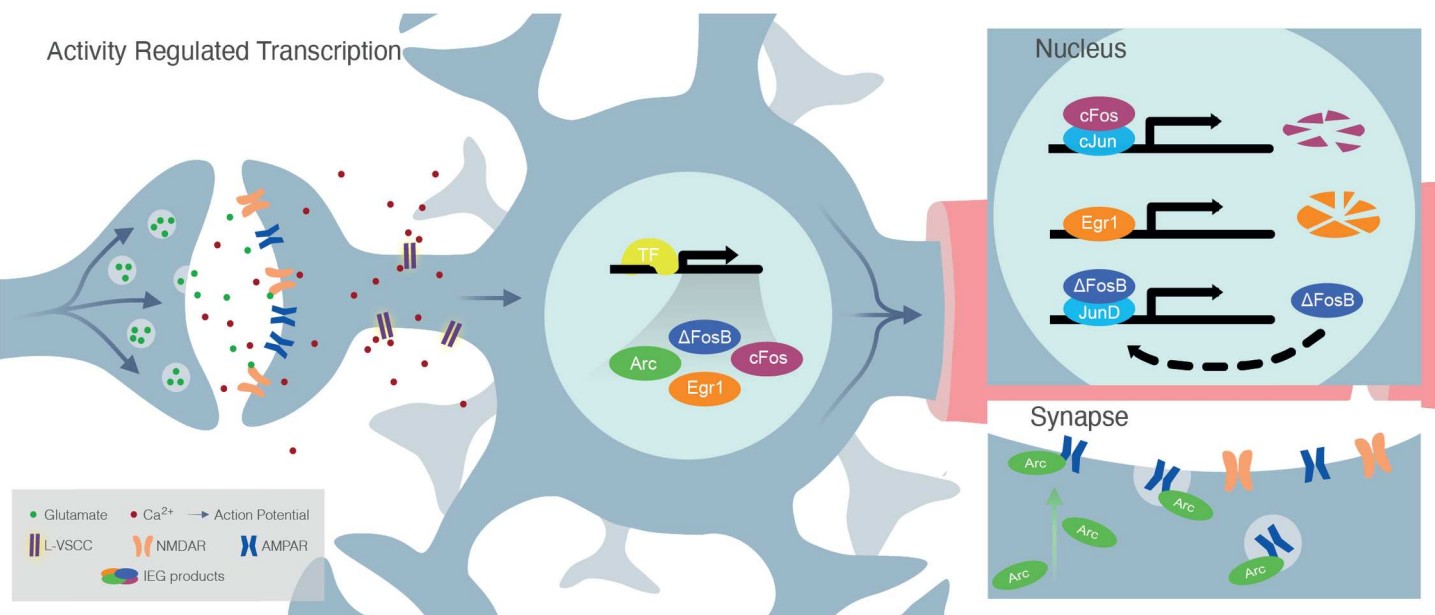

**Fig 1. Mechanisms of IEG activation.** *A)* Action potentials induced by glutamate signaling result in membrane depolarization, which in turn results in opening of L-type voltage calcium channels (LVCCs). The resulting $Ca^{2+}$ influx induces calcium-dependent signaling pathways. These cascades further result in the recruitment of existing transcription factor, such as CREB, which in turn lead to the expression of IEGs. Once transcribed, IEGs act as *B)* transcription factors in the nucleus or *C)* regulators of synaptic plasticity at the synapse as, for example, post-synaptic proteins. *B)* The transcription factors of the *Fos* family bind to a transcription factor of the *Jun* family to form the AP1 complex, whereas Egr1 acts independently. Egr1 and cFos are transiently expressed, whereas ΔFosB accumulates over time in the nucleus. *C)* Arc acts at the post-synaptic density by reducing the number of surface AMPA receptors. Therefore, increased Arc expression results in reduced synaptic strength by AMPA receptor endocytosis.

IEG expression after an acute stress challenge, further amplified by exposure to additional hits in life, in line with the multiple-hit concept of vulnerability [27].

## Methods

Search strategy, protocol and risk of bias assessment of the present review were performed in line with SYRCLE (Systematic Review Center for Laboratory animal Experimentation) guidelines [28–30]. We adhered to the PRISMA checklist for reporting [31] (Supporting Information). The protocol (S1.1 in S1 File) and the PRISMA checklist are openly accessible at https://osf.io/qkyvd/.

### Study selection and data extraction

We conducted a systematic literature search with the search engines PubMed and Embase on the 3rd of April 2019 to select experiments investigating differences in IEG expression between control and ELA exposed rodents. The terms *'mice and rats'* and *'postnatal ELA'* were used to construct the search string (S1.2 in S1 File). For the purpose of this review, ELA was defined as models altering maternal care. We included the ELA models of maternal separation and deprivation, isolation, limited bedding and nesting, as well as licking and grooming. Study selection was performed in Rayyan [32] in alphabetical order and any disagreements between investigators were resolved by discussion until unison was reached. An overview of the study selection procedure is displayed in Fig 2.

A complete list of final inclusion and exclusion criteria can be found in the protocol (S1.1 in S1 File). First, titles and abstracts were screened by at least three blinded investigators (HS, VB, EK, DvN, LvM) for the following exclusion criteria: 1) not a primary experimental

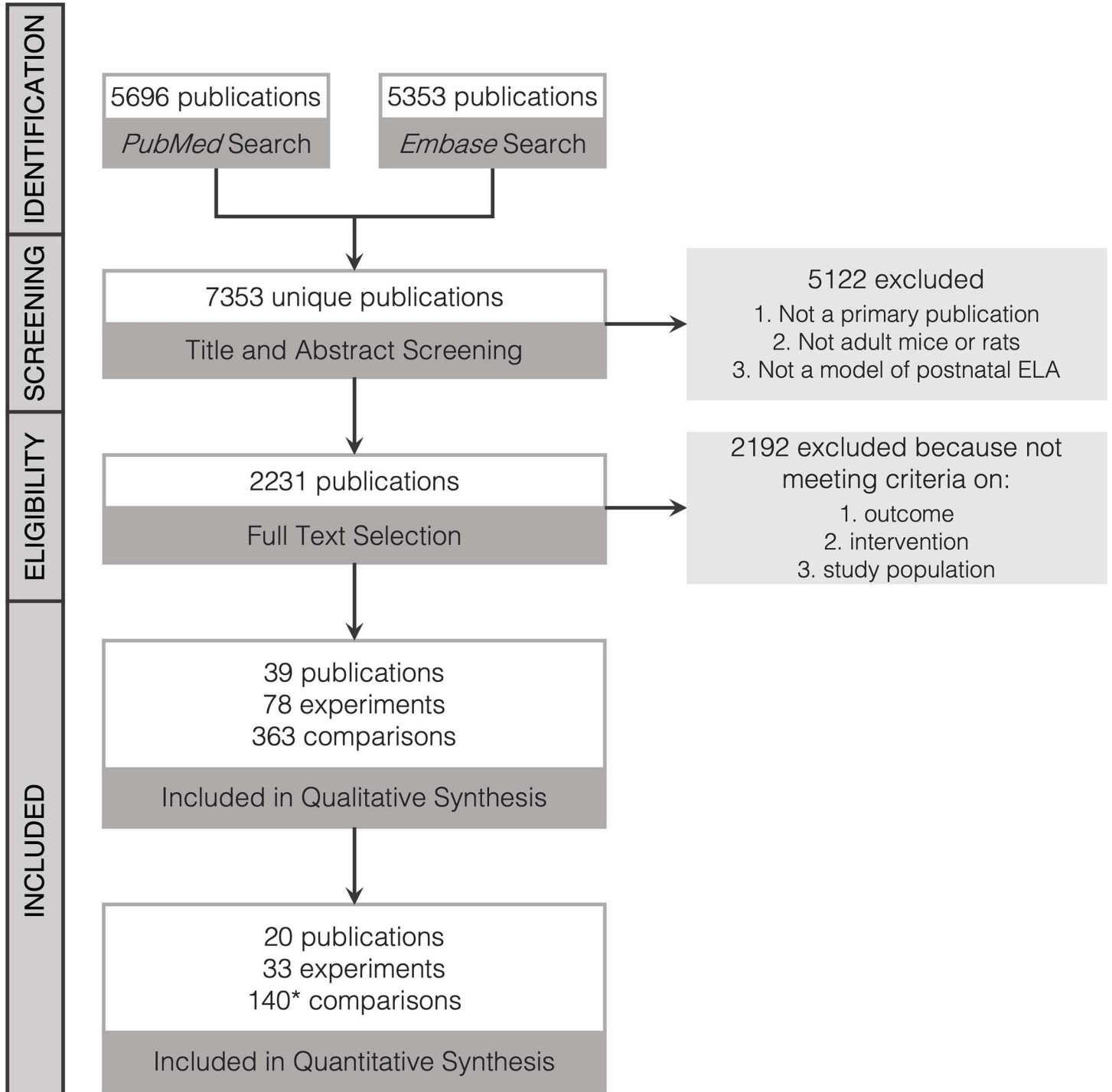

**Fig 2. Flow-chart of study selection process.** ^ = not included in pre-specified inclusion/exclusion criteria (S1.1 in S1 File).

publication, 2) not using adult (>8 weeks) mice or rats which are younger than 1 year, 3) not using a postnatal model of ELA as specified in S1.1 in S1 File. Eligibility was then determined by full-text screening of the remaining studies by at least two blinded investigators (HS, EK, LvM), with a random subset screening performed by a fourth (VB), blinded investigator to

confirm agreement. Publications were deemed non-eligible based on the following criteria: 1) not measuring an IEG product in the brain, 2) deviation from *a priori* determined criteria concerning the background of the animals, interventions, or outcomes, 3) control and experimental groups differed at more aspects than just ELA exposure. Lastly, reference sections of eligible publications were screened for articles missed by the search string, but none were added through this procedure.

Data from eligible studies were extracted into a combined dataset using *a priori* determined sets of variables to comprehensively capture experimental design, methods and results with minimal subjectivity (S1.3 in S1 File). Differently from the original protocol, we extracted also measurements without acute stress to have an appropriate control, baseline condition. Outcome data for each comparison (*i.e.* group-based mean and variance) were extracted in the following order of preference: 1) from numbers provided in the text or tables; 2) from graphs by using WebPlotDigitizer (v4.3 [33]; or 3) from statistical test results. A comparison is defined as the difference in expression of a specific IEG in a specific brain area at rest or after acute stress exposure in ELA-exposed animals and controls. To compare the results on a systematic review level, we performed an independent samples *t*-tests on the extracted summary statistics. The results were interpreted dichotomously as significant / not significant, with $p < 0.05$ used as a criterion. We chose this approach to equalize the statistical method used for analysis across publications.

## Meta-analysis

**Data selection.** We performed a meta-analysis on outcomes that were assessed by at least three independent comparisons (*i.e.*, at least one comparison from three independent publications). During analysis coding, the investigators were blinded to the outcome by randomly multiplying half of the effect sizes by -1.

To account for potential sex differences, we planned to perform separate meta-analyses for males and females. However, only few comparisons were reported for female rodents, and their study designs were strongly heterogeneous. We therefore restricted our quantitative synthesis to outcomes from male rodents, with female data being evaluated qualitatively only. Furthermore, only comparisons using either IHC, immunocytochemistry (ICC), or ISH to quantify IEG expression were included on the meta-analytic level. While both methods differ in the type of molecule being assessed, quantification and analysis procedures largely overlap. To confirm this, we investigated whether the choice of quantification method affects the outcome. PCR based methods and western blots were evaluated qualitatively only.

Based on the aforementioned threshold and restrictions, the meta-analysis was performed on comparisons of cFos expression in the amygdala, thalamus, hippocampal formation, hypothalamus, prefrontal cortex and midbrain at rest and after acute stress experiences. Smaller subregions were grouped into larger structures (S1.4 in S1 File) in line with the Allen Mouse Brain Atlas (©2004, Allen Institute for Brain Science) to allow for comparisons between studies.

**Statistical analysis.** For comparisons included in the meta-analysis, we calculated the standardized mean difference Hedge's *g* as a measure of effect size. If only the standard error of the mean (SEM) was reported, the standard deviation (SD) was calculated as $SEM^* \sqrt{n}$, where *n* = the number of animals per group. If the total number of animals was reported, this was distributed equally across groups. If the number of animals was reported as a range (e.g.6-8 animals/group), we used the mean (e.g. 7 animals/group). If the same control group was used as control of multiple experimental groups (e.g. different ELA models), the sample size of the control group was divided by the number of experimental groups and the adjusted sample

size was used for the calculation of the effect size [34]. Heterogeneity was assessed with Cochran's *Q*-test [35]. Influential outliers were determined in accordance with Viechtbauer and Cheung [36] and removed from quantitative synthesis. Of such comparisons, we explored whether elements of the experimental design could explain the deviation of these comparisons from the mean.

A three-level mixed-effects model was built to capture variance not only between publications (Level 1), but also between experiments (Level 2) and outcomes (Level 3), thereby taking into account the statistical dependency of outcomes acquired from the same animals within the same publication [37–39]. Moderators of the multilevel model were i) presence of an acute stress challenge, ii) presence of additional hits and iii) brain area.

We tested whether ELA effect sizes at rest or after acute stress challenges are significantly different from zero to understand the effects of ELA on *cFos* expression under each of these conditions. Subsequently, a subgroup analysis was performed to investigate whether the effects are moderated by the experience of multiple negative life experiences (additional hits). The presence of additional hits was classified with previously determined criteria [16]. Finally, we explored the effects of type of acute stressor (i.e. *mild* versus *severe*, S1.4 in S1 File), novelty of stress experience, and brain region using subgroup analyses.

**Bias assessment and sensitivity analyses.** We followed SYRCLE guidelines on risk of bias assessment, with items not reported being coded as 'unclear' [30]. To detect publication bias, funnel plot asymmetries for each outcome variable were evaluated [30]. Due to the uneven frequency of the number of studies, we performed sensitivity analyses (rather than subgroup as specified in the protocol) on the type of ELA model, and difference between mRNA and protein. Since these analyses were not initially included, the results were only qualitatively assessed and were in line with the interpretation of the main results. All analyses can be found at our repository (https://osf.io/qkyvd/).

**Software.** All analyses were performed in R (v3.6.1; [40]). The following R packages were used: etaphor (v 2.1.0; [41]), tidyverse (v1.2.1; [42]). Data are presented as the standardized mean difference Hedge's *g* and standard error of the mean ($g[\pm SEM]$). The significance level α was set to 0.05. Multiple testing correction on the planned analysis was performed using the Holm-Bonferroni method [43]. The code for analysis is openly accessible at https://osf.io/qkyvd/.

## Results

### Study selection and characteristics

A total of 1019 animals reported in 39 publications were included in the review. The animals were predominantly male (72.5%); rats (76.3%) were used more often than mice; and protein (77.4%) rather than mRNA was more frequently assessed as outcome. The IEG *cFos* was investigated in the majority of studies (88.7%), and maternal separation was the most frequently used ELA model (90.6%). Fig 3 shows a graphical overview of the study characteristics.

### Research synthesis

**Systematic review of *cFos* and ELA.** A total of 31 publications reported cFos expression in control and ELA animals (Table 1). IEG expression was reported to be significantly affected by ELA in 72 (45.8%) comparisons, of which 33 (59.6%) displayed upregulation and 39 comparisons (54.2%) reported downregulation.

Overall, of the 322 comparisons within these studies, 140 comparisons ($n_{pub} = 20$) qualified for further meta-analysis in male rodents after removal of influential outliers ($n_{comp} = 1$); these are analyzed quantitatively in the following section. No element of the experimental design

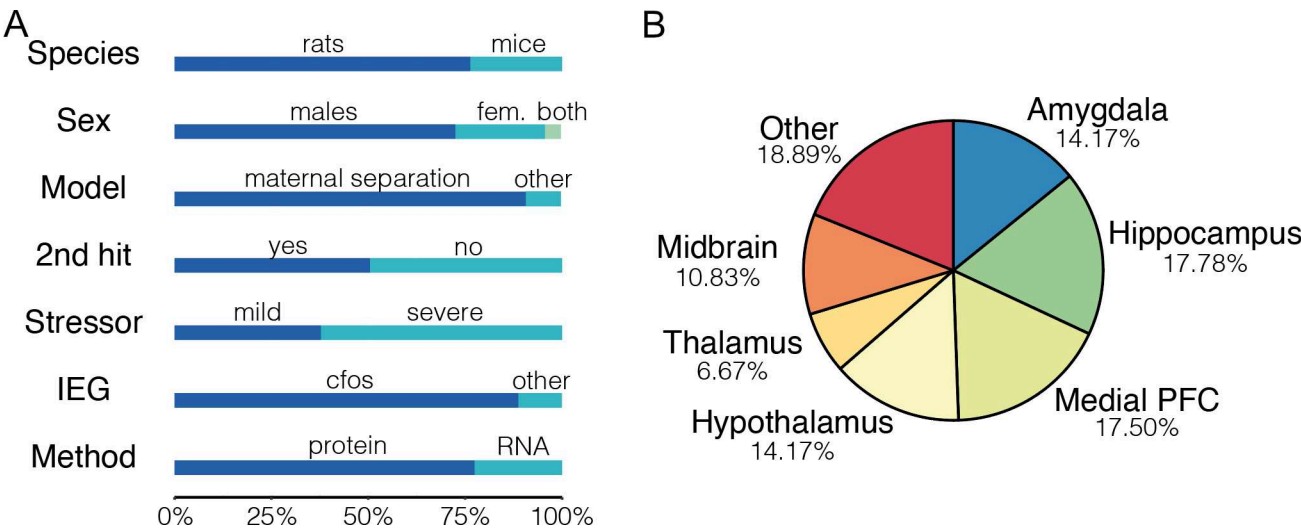

**Fig 3.** A) Study characteristics and B) Investigated brain areas reported as percentage of experiments. *Fem* = females.

pointed towards a biological origin of the outlying value, nor was its publication published in a predatory journal [75]. Comparisons were excluded from quantitative review because of brain area ($n_{comp}$ = 40), acute stressor type ($n_{comp}$ = 49; S2.4.3 in S1 File) or unspecified or pooled sex ($n_{comp}$ = 15). The excluded comparisons are subject to a qualitative review in the Supporting Information.

Given fundamental biological differences between males and females [76], we *a priori* chose to evaluate female cFos data separately from males'. However, only ten publications reported on cFos expression in female rodents ($n_{comp}$ = 77). Given the limited number of studies, with variable designs, we had to abandon the separate meta-analytical evaluation of female rodents. Qualitatively, the majority of the studies with females found no significant differences between cFos levels of ELA versus controls at rest or after an acute stress challenge ($n_{comp}$ = 55 [51, 54, 55, 57, 62]). A more detailed description is supplied in the Supporting Information.

**Systematic review of ELA and other IEGs.** We here only summarize the main findings on IEGs other than cFos. In general, the number of studies on these IEGs compared to cFos was very limited. For a more elaborate description and discussion we refer to the Supporting Information.

*Arc* is a post-synaptic protein, which plays an essential role in regulating the homeostatic scaling of AMPA receptors, thereby directly modifying plasticity at the synapse [77]. *Arc* expression was investigated in only five publications under varying conditions in male and female mice and rats (see Table 2 and Supporting Information).

Early-growth response (Egr) proteins are a family of transcription factors with a zinc-finger motif, which allows all Egr factors to connect to identical DNA binding sites [82]. We identified only three studies investigating Egr expression after ELA exposure at rest (Table 3 and Supporting Information); specifically, one investigated Egr-1 [83], another investigated Egr-4 only [80], and one other investigated Egr-2 and Egr-4 [79].

*FosB* is an IEG of the *Fos* family, and—similarly to *cFos*—it binds to members of the *Jun* family to form the AP1 transcription factor [84]. Of particular interest in stress research is its isoform *ΔFosB*, whose extended half-life makes it an exceptional marker for chronic stress [84]. Three publications reporting on the expression of *ΔFosB* at rest in ELA and control animals were identified (see Table 4 and Supporting Information).

**Table 1. Overview of study designs and findings of reviewed publications reporting on cFos expression in ELA and control animals.**

| Author (Year) | MA | Model (PNDs) | Species | Sex | Exp. Design details | AS | Effect | Area(s) |
|---|---|---|---|---|---|---|---|---|
| Auth (2018) [44] | | MS (2–15) | Mouse | F | Dark-light box | ✓ | ↔ | BLA, LA, CEA, PVN |
| | | | | | Open-field test | ✓ | ↔ | BLA, LA, CEA, PVN, dlPAG, vlPAG |
| | | | | | Two independent naïve cohorts | ✗ | ↑ | dlPAG |
| | | | | | | | ↔ | BLA, CEA, PVN, vlPAG, LA |
| | | | | | | ✗ | ↑ | LA |
| | | | | | | | ↔ | BLA, LA, CEA, PVN |
| Banqueri (2018) [45] | | MS (1–10) | Rat | F | Morris water maze | ✓ | ↑ | CA1, DG |
| | | | | | | | ↓ | ACA |
| | | | | | | | ↔ | avTN, amTN, IL, PL |
| | | MS (1–21) | Rat | F | Morris water maze | ✓ | ↑ | DG |
| | | | | | | | ↓ | IL, PL, ACA |
| | | | | | | | ↔ | avTN, amTN |
| Benner (2014) [46] | | MS (2–15) | Mouse | M | Competitive dominance task | ✓ | ↑ | BLA |
| | | | | | | | ↓ | CA1 |
| | | | | | | | ↔ | ACA, CEA, DG, IL, PL |
| Chung (2007) [47] | ✓ | MS (2–14) | Rat | M | Colorectal distension | ✓ | ↑ | ACA |
| | | | | | | | ↔ | CEA, cmTN, PAG, PVT, vmHN |
| | | | | | - | ✗ | ↑ | ACA |
| | | | | | | | ↔ | CEA, cmTN, PAG, PVT, vmHN |
| Clarke (2013) [48] | ✓ | MS (10–11) | Rat | M | Small litter (12 pups); Restraint stress | ✓ | ↑ | mPPVN |
| | | | | | | | ↔ | vBNST, MGPVN, lPPVN, dPPVN |
| | | | | | Small litter | ✗ | ↓ | dPPVN |
| | | | | | | | ↔ | mPPVN, MGPVN, lPPVN, vBNST |
| | | | | | Large litter (20 pups); Restraint stress | ✓ | ↔ | mPPVN, MGPVN, lPPVN, dPPVN, vBNST |
| | | | | | Large litter | ✗ | ↔ | mPPVN, MGPVN, lPPVN, dPPVN, vBNST |
| Cohen (2013) [49] | ✓ | LBN (2–21) | Mouse | M | Novel environment | ✓ | ↔ | BLA |
| Daskalakis (2014) [50] | | MS (3–5) | Rat | M | MS pups remained in HC; re-exposure to fearful context | ✓ | ↑ | MEA |
| | | | | | | | ↔ | BLA, CEA |
| | | | | | MS pups placed in NC; re-exposure to fearful context | ✓ | ↑ | BLA, MEA |
| | | | | | | | ↔ | CEA |
| Desbonnet (2008) [51] | ✓ | MS (2–14) | Rat | M | Forced swim test | ✓ | ↔ | PVT, CEA, PVN, BNST, DG |
| | | | | | - | ✗ | ↔ | PVT, CEA, PVN, BNST, DG |
| | | | | F | Forced swim test | ✓ | ↔ | PVT, CEA, PVN, BNST, DG |
| | | | | | - | ✗ | ↔ | PVT, CEA, PVN, BNST, DG |
| Felice (2014) [52] | ✓ | MS (2–12) | Rat | M | Open-field test | ✓ | ↔ | BLA, CEA, rostral & caudal ACA, IL, PL |
| | | | | | Colorectal distension | ✓ | ↑ | rostral & caudal ACA, IL, PL |
| | | | | | | | ↔ | BLA, CEA |
| | | | | | - | ✗ | ↔ | BLA, CEA, rostral & caudal ACA, IL, PL |

*(Continued)*

**Table 1.** (Continued)

| Author (Year) | MA | Model (PNDs) | Species | Sex | Exp. Design details | AS | Effect | Area(s) |
|---|---|---|---|---|---|---|---|---|
| Gardner (2005) [53] | | MS (2–14) | Rat | M | Social defeat paradigm; cFos counts summed across 4 slices | ✔ | ↔ | DRN |
| | | | | | | ✘ | ↔ | DRN |
| | | Handling (2–14) | Rat | M | Social defeat paradigm; cFos counts summed across 4 slices | ✔ | ↔ | DRN |
| | | | | | | ✘ | ↔ | DRN |
| Gaszner (2009) [54] | | MS (8–14) | Rat | M | Restraint stress | ✔ | ↔ | EW |
| | | | | | - | ✘ | ↔ | EW |
| | | | | F | Restraint stress | ✔ | ↔ | EW |
| | | | | | - | ✘ | ↔ | EW |
| | | Handling (8–14) | Rat | M | Restraint stress | ✔ | ↔ | EW |
| | | | | | - | ✘ | ↔ | EW |
| | | | | F | Restraint stress | ✔ | ↑ | EW |
| | | | | | - | ✘ | ↔ | EW |
| Genest (2004) [55] | ✔ | MS (3–12) | Rat | M | Novel environment | ✔ | ↑ | PVN |
| | | | | F | Novel environment | ✔ | ↔ | PVN |
| Hidaka (2018) [56] | | MS (2–14) | Mouse | M | Three chamber test | ✔ | ↔ | ACA, IL, PL |
| James (2014) [57] | ✔ | MS (2–14) | Rat | M | Restraint stress | ✔ | ↓ | mPPVN |
| | | | | | | | ↔ | PVT |
| | | | | F | Restraint stress | ✔ | ↔ | mPPVN, PVT |
| Loi (2017) [58] | | MS (3–4) | Rat | M | Rodent Iowa gambling task | ✔ | ↓ | rCA1, rCA3, leAI, leIL |
| | | | | | | | ↔ | r&leDG, r&leACA, r&lePL, le CA1, le CA3, r&le dlSX, r&le mlSX, r&le AI, r&le NAcc Shell&Core, rIL, r&le vOFC, r&le mOFC, r&le cOFC |
| Menard (2004) [59] | ✔ | LG | Rat | M | Shock-probe burial task with electrified probe | ✔ | ↓ | dlSX, vlSX, vSUB, dPAG, vPAG |
| | | | | | | | ↔ | vDG, dDG, mSX, CA1, CA3, aHN, CEA, BLA, lC, NAcc shell |
| O'Leary (2014) [60] | | MS (1–14) | Mouse | F | Restraint stress | ✔ | ↓ | dDG, vCA3 |
| | | | | | | | ↔ | dCA1, dCA2, PVN, dCA3, vdG, NAcc, VTA, IL, PL, ACA, LA, BLA, CEA, DRN |
| Ren (2007) [61] | | MS (2–21) | Rat | M | Colorectal distension | ✔ | ↔ | DRN |
| Renard (2010) [62] | ✔ | MS (1–21) | Rat | M | Perfusion 24h after last day of chronic variable stress | ✘ | ↔ | mPPVN |
| | | | | F | | | ↔ | mPPVN |
| | | | | M | - | ✘ | ↔ | mPPVN |
| | | | | F | | | ↔ | mPPVN |
| Rincel (2016) [63] | ✔ | MS (2–14) | Rat | M | Open-field test | ✔ | ↓ | PVN |
| Rivarola (2008) [64] | | MS (1–21) | Rat | F | Perfusion 24h after last day of chronic variable stress | ✘ | ↑ | adTN |
| | | | | | - | ✘ | ↑ | adTN |
| Rivarola (2009) [65] | | MS (1–21) | Rat | F | Perfusion 24h after last day of chronic variable stress | ✘ | ↑ | RSP |
| | | | | | | | ↔ | adTN, MMN |
| | | | | | - | ✘ | | adTN, RSP |

*(Continued)*

**Table 1.** (Continued)

| Author (Year) | MA | Model (PNDs) | Species | Sex | Exp. Design details | AS | Effect | Area(s) |
|---|---|---|---|---|---|---|---|---|
| | | | | | | | ↔ | MMN |
| Shin (2018) [66] | ✓ | MS (1–14) | Mouse | M | Social interaction after 1d social isolation | ✓ | ↑ | lSX, VTA |
| | | | | | | | ↔ | mPfC, NAcc, vPAL, AHA, VH |
| | | | | | - | ✗ | ↔ | lSX, VTA, mPfC, NAcc, vPAL, AHA, VH |
| Tenorio-Lopes (2017) [67] | ✓ | MS (3–12) | Rat | M | Novel Environment | ✓ | ↔ | BLA, CEA, MEA, DMH, PVN |
| Troakes (2009) [68] | ✓ | MS (5–21) | Rat | M | Elevated plus maze | ✓ | ↓ | PIR |
| | | | | | | | ↔ | ACA, SSb, lSX, PVN, CEA, MEA, dCA1, vCA1, dCA2, vCA2, dCA3, vCA3, dDG, vDG, CP, DRN, Pontine region, CB |
| | | | | | - | ✗ | ↔ | ACA, SSb, PIR, lSX, PVN, CEA, MEA, dCA1, vCA1, dCA2, vCA2, dCA3, vCA3, dDG, vDG, CP, DRN, Pontine region, CB |
| Trujillo (2016) [69] | ✓ | MS (1–21) | Rat | M | Perfusion 24h after last day of chronic variable stress | ✗ | ↑ | MEA |
| | | | | | | | ↔ | CA1, CA2, CA3, PVN |
| | | | | | - | ✗ | ↑ | CA1, CA2, CA3, MEA |
| | | | | | | | ↔ | PVN |
| van Hasselt (2012) [70] | | LG | Rat | P | Rodent Iowa gambling task; results reported as correlation with %LG | ✓ | ↑* | NAcc Shell, AI |
| | | | | | | | ↔* | mOFC, vOFC, lOFC, ACA, PL, IL, dlSTR, dmStR, NAcc Core, CEA, BLA, DG, CA1 |
| Vivinetto (2013) [71] | ✓ | MS (1–21) | Rat | M | Foot shock in step-down inhibitory avoidance task | ✓ | ↔ | CA1, CA3, DG |
| Yajima (2018) [72] | | MS (2–14) | Mouse | M | - | ✗ | - | HPF |
| Zhang (2009) [73] | ✓ | MS (2–14) | Rat | M | Colorectal distension | ✓ | ↑ | cmTN |
| | | | | | | | ↔ | ACA, vplTN, PVT |
| | | | | | - | ✗ | ↑ | vplTN |
| | | | | | | | ↔ | ACA, cmTN, PVT |
| Zhao (2013) [74] | ✓ | MS (2–14) | Rat | M | Chinese language publication | ✗ | ↑ | PVN |

Header: *MA*–whether some or all comparisons from this study are included in the meta-analysis (✓) or on systematic review level only; *Model(PNDs)*–which ELA model (MS–maternal separation, LBN–limited bedding and nesting, LG–licking and grooming) was applied during which postnatal days (PNDs); *Sex*–animals were female (F) or male (M); *Exp. design details*–indicates how experiments (nests) differed, if–then rest/no manipulation; *AS*–if acute stress challenge was present (✓) or not (✗); *Effect*–if ELA significantly increased (↑), decreased (↓) or did not alter (↔) IEG expression as based on independent *t*-tests

* = t-test could not be performed and effects are shown as reported in the original publication; *Areas*–brain areas as identified in publication, with position (lowercase, if identified) and area acronym as follows

Area acronyms (in alphabetical order): ACA–anterior cingulate area; AHA–anterior hypothalamic nucleus; AI–agranular insular cortex; BLA–basolateral amygdala; BNST–bed nuclei of the stria terminalis; CB–cerebellum; CEA–central amygdala; CP–caudate putamen; CTX–cortex; DG–dentate gyrus; DRN–dorsal raphe nucleus; EW–Edinger-Westphal nucleus; HN–hypothalamic nucleus; HPF–hippocampal formation; IL–infralimbic area; DMH–dorsomedial hypothalamic nucleus; LA–lateral amygdala; LC–locus coeruleus; MEA–medial amygdala; MGPVN–magnocellular part of the PVN; MMN–mammillary nucleus; NAcc–nucleus accumbens; OFC–orbital-frontal cortex; PAG–periaqueductal gray; PAL–Pallidum; PFC–prefrontal cortex; PIR–Piriform cortex; PL–prelimbic area; PPVN–parvocellular part of the PVN; PVN–paraventricular nucleus of the hypothalamus; PVT–paraventricular nucleus of the thalamus; RSP–retrosplenial cortex; SSb–somatosensory barrel cortex; STR–striatum; SUB–subiculum; SX–septum; TN–thalamic nucleus; VH–ventral hypothalamic nucleus; VTA–ventral tegmental area.

Position: a–anterior; c–central; d–dorsal; l–lateral; le–left; m–medial; p–posterior; r–right; v–ventral.

**Table 2. Overview of study designs and findings of reviewed publications reporting on Arc expression in ELA and control animals.**

| Author (Year) | Model (PNDs) | Species | Sex | Exp. design details | AS | Effect | Area(s) |
|---|---|---|---|---|---|---|---|
| Benekareddy (2010) [78] | MS (2–14) | Rat | M | - | ✖ | ↔ | mPFC |
| Benner (2014) [46] | MS (2–15) | Mouse | M | Competitive dominance task | ✔ | ↔ | ACA, BLA, CEA, CA1, DG, IL, PL |
| McGregor (2018) [79] | MS (2–14) | Rat | M | Juvenile restraint stress | ✖ | ↑* | dSTR |
| | | | | - | ✖ | ↑* | dSTR |
| Rincel (2019) [80] | MS (2–14) | Mouse | M | - | ✖ | ↓* | mPFC |
| | | | F | - | ✖ | ↑* | mPFC |
| Solas (2010) [81] | MS (2–21) | Rat | M | - | ✖ | ↓ | CA1, CA3, DG |

Header: *Model(PNDs)*–which ELA model (MS–maternal separation) was applied during which postnatal days (PNDs); *Sex*–animals were female (F) or male (M) or not specified (NS); *Exp. design details*–indicates how experiments (nests) differed, if–then rest/no manipulation; *AS*–if acute stress challenge as present (✔) or not (✖); *Effect*–if ELA significantly increase (↑), decreased (↓) or did not alter (↔) IEG expression as based on independent *t*-tests

\* = t-test could not be performed and effects are shown as reported in the original publication; *Areas*–brain areas as identified in publication, with area acronym as follows

Area acronyms (in alphabetical order): ACA–anterior cingulate area; BLA–basolateral amygdala; CEA–central amygdala; DG–dentate gyrus; IL–infralimbic area; mPFC–medial prefrontal cortex; PL–prelimbic area; dSTR–dorsal striatum.

**Meta-analysis of cFos in male rodents.** For cFos, our survey yielded sufficient data to carry out a meta-analysis, next to the systematic review. In comparison to control animals, rodents with a history of ELA displayed significantly increased cFos levels at rest ($g[SEM]$ = 0.421[±0.18], $t$ = 2.35, $p_{adj}$ = 0.041), but not after acute stress exposure ($g[SEM]$ = 0.133 [±0.166], $t$ = 0.805, $p_{adj}$ = 0.422; Fig 4A). To gain a deeper understanding of these findings, we performed subgroup analyses to investigate the experience of additional hits, i.e. an additional negative life event. Of note, the control and experimental groups always differed only in the presence/absence of ELA. Therefore, in the 'additional hits' comparisons, both control and ELA animals experienced multiple negative life events. This was important for cFos expression after acute stress, where the effects of ELA were pronounced only in synergy with additional hits (Fig 4B, <u>acute</u>no hit: $g[SEM]$ = -0.193[±0.135], $z$ = -1.436, $p_{adj}$ = 0.151; <u>acute</u>mult hits: $g[SEM]$

**Table 3. Overview of study designs and findings of reviewed publications reporting on expression of the Egr-family in ELA and control animals.**

| Author (Year) | Model (PNDs) | Species | IEGs | Sex | Exp. design details | AS | Effect | Area(s) |
|---|---|---|---|---|---|---|---|---|
| McGregor (2018) [79] | MS (2–14) | Rat | Egr-2 | M | Juvenile restraint stress | ✖ | ↔* | dSTR |
| | | | | | - | ✖ | ↑* | dSTR |
| | | | Egr-4 | M | Juvenile restraint stress | ✖ | ↑* | dSTR |
| | | | | | - | ✖ | ↑* | dSTR |
| Navailles (2010) [83] | MS (2–15) | Mouse | Egr-1 | M | Balb/c strain | ✖ | ↓ | CTX |
| | | | | | | | ↔ | DG, CA1, CA2, CA3 |
| | | | | | C57BL/6 strain | ✖ | ↔ | CTX |
| Rincel (2019) [80] | MS (2–14) | Mouse | Egr-4 | M | - | ✖ | ↓* | mPFC |
| | | | | F | - | ✖ | ↑* | mPFC |

Header: *Model(PNDs)*–which ELA model (MS–maternal separation) was applied during which postnatal days (PNDs); *Sex*–animals were female (F) or male (M) or not specified (NS); *Exp. design details*–indicates how experiments (nests) differed, if–then rest/no manipulation; *AS*–if acute stress challenge as present (✔) or not (✖); *Effect*–if ELA significantly increase (↑), decreased (↓) or did not alter (↔) IEG expression

\* = t-test could not be performed and effects are shown as reported in the original publication; *Areas*–brain areas as identified in publication, with area acronym as follows

Area acronyms (in alphabetical order): CTX–cortex; DG–dentate gyrus; dSTR–dorsal striatum; mPFC–medial prefrontal cortex.

**Table 4. Overview of study designs and findings of reviewed publications reporting on ΔFosB expression in ELA and control animals.**

| Author (Year) | Model (PNDs) | Species | Sex | Exp. design details | AS | Effect | Area(s) |
|---|---|---|---|---|---|---|---|
| Kim (2015) [85] | MS (1–14) | Rat | F | - | ✖ | ↓ | NAcc |
| Lippmann (2007) [86] | MS (2–14) | Rat | M | - | ✖ | ↔* | CTX, NAcc, STR |
| | Handling (2–14) | Rat | M | - | ✖ | ↔* | CTX, NAcc, STR |
| Wang (2016) [87] | MS (1–15) | Rat | NS | - | ✖ | ↑ | mPFC |

Header: *Model(PNDs)*–which ELA model (MS–maternal separation) was applied during which postnatal days (PNDs); *Sex*–animals were female (F) or male (M) or not specified (NS); *Exp. design details*–indicates how experiments (nests) differed, if–then rest/no manipulation; *AS*–if acute stress challenge as present (✔) or not (✖); *Effect*–if ELA significantly increase (↑), decreased (↓) or did not alter (↔) IEG expression

\* = t-test could not be performed and effects are shown as reported in the original publication; *Areas*–brain areas as identified in publication, with area acronym as follows

Area acronyms (in alphabetical order): CTX–cortex; mPFC–medial prefrontal cortex; NAcc–nucleus accumbens; STR–striatum.

= 0.442[±0.159], z = 2.784, p = 0.016; <u>at rest</u>$_{no\ hit}$: g[SEM] = 0.475 [±0.16], z = 2.976, p < .012; <u>at rest</u>$_{mult\ hits}$: g[SEM] = 0.344[±0.153], z = 2.253, p = 0.049; the analyses were conducted comparing the effect size between control and ELA animals against 0). Lastly, we performed an exploratory analysis to investigate potential interactions with acute stressor severity on the

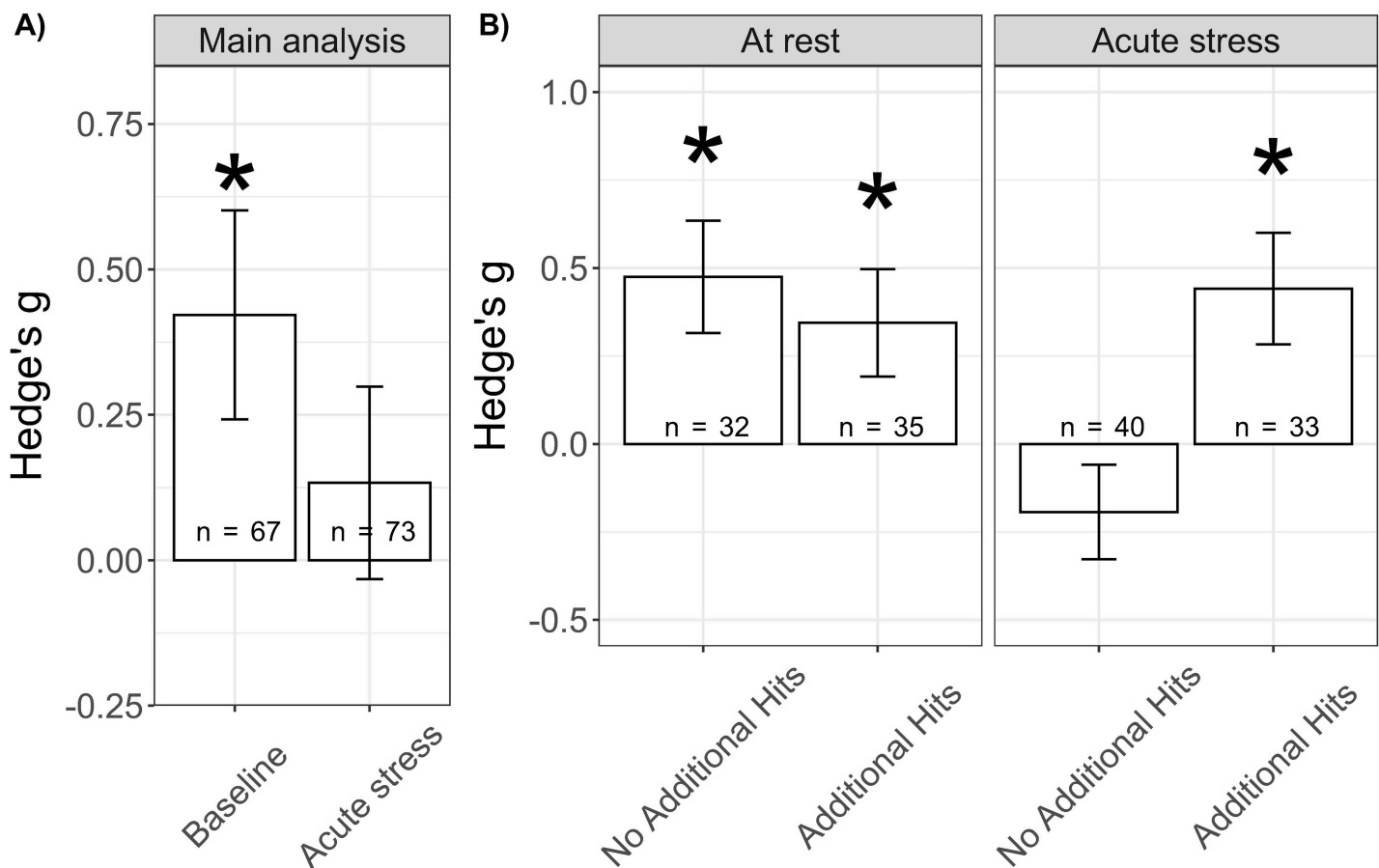

**Fig 4. Main and subgroup analyses.** A) Effects of ELA on cFos expression in male rodents at rest and after an acute stressor. B) Subgroup analysis for absence (No Additional Hits) or presence (Additional Hits) of additional negative life events. Of note, control and experimental animals always differed only in the presence/absence of ELA. Therefore, in the 'Additional Hits' comparison, also control animal experienced the additional negative life events. * p < 0.05.H.

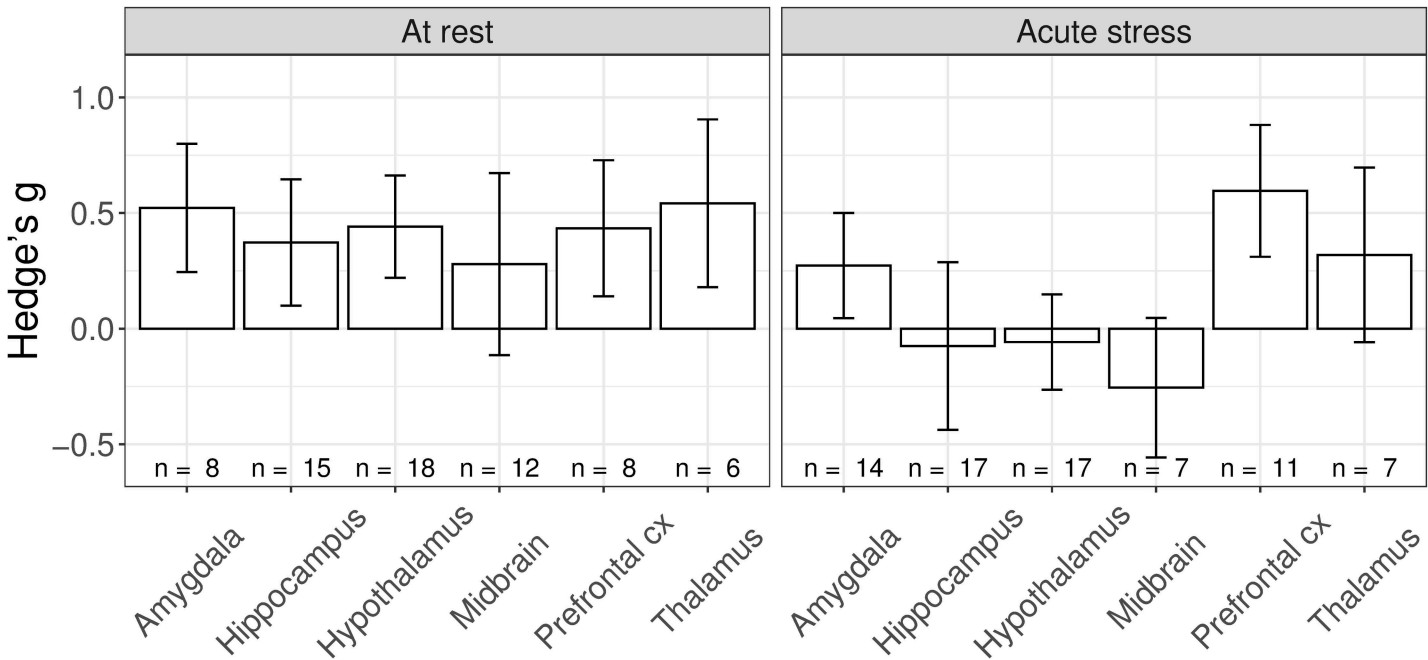

**Fig 5. Effects of ELA on *cFos* expression across brain areas at rest and after an acute stressor.**

effect sizes. For the categorization of acute stressor severity, please see Supporting Information S1.4 in S1 File. Acute stressor severity was not a significant moderator ($Q_M(3) = 4.35$, $p = 0.226$). Of note, no publication investigated cFos expression after a mild acute stressor in animals that experienced additional hits ($n_{comp} = 0$). Of the 20 publications included in the meta-analysis, only two did not use maternal separation as an ELA model [49, 59]. Nevertheless, these studies adhere to the above findings with no significant differences found after acute stress in the areas meta-analytically investigated. The findings of our main analysis do not confirm our hypothesis that cFos expression is higher in ELA animals compared to control particularly after acute stress; rather, the results indicate that cFos expression is increased after ELA already at baseline, i.e. at rest. Moreover, the results highlight the relevance of including the presence of additional hits in the analysis.

Next, we tested whether the effects of ELA on cFos expression differed across brain regions important for the stress reaction (Fig 5), when only considering those datasets with sufficient observations ($n_{publications} > 3$). Brain region was not a significant moderator ($Q_M(12) = 13.908$, $p = 0.307$) of the effects of ELA on cFos expression. Exploratory subgroup analysis suggests that at rest all brain areas show a comparable increase in cFos expression. After an acute stress challenge, the effects appeared more variable across brain areas than at rest. We then performed an additional exploratory analysis to investigate whether brain areas after acute stress differed after ELA with / without the experience of additional hits. The results of this analysis suggested that the prefrontal cortex may be specifically affected; however, since this effect was supported by those studies unevenly represented in the funnel plot, these results may not be reliable due to presumed publication bias.

Despite significant contribution of the moderators ($Q_M(23) = 40.089$, $p = 0.015$), residual heterogeneity between studies remained significant ($Q_E(117) = 167.95$, $p = 0.001$). Study of the distribution of variance showed that remaining variance is mainly attributable to differences between experiments (Level 2) and not to differences within experiments (Level 3).

Concerning potential bias, while reporting risk of bias was incomplete in all publications (S2.1a in S1 File), 46% of studies reported adequate randomization and blinding procedures ($n_{pub}$ = 10). Visual assessment of the funnel plot for the studies qualifying for quantitative synthesis suggests the presence of publication bias (S2.1b in S1 File), which was also supported by a significant Eggers' test (z = 4.6903, p < .0001). We identified two studies [52, 69] which were mainly responsible for the bias.

## Discussion

In this review, we synthesized the evidence of 39 publications investigating the effects of ELA on IEG expression in mice and rats. Due to low number of animals used in preclinical research, studies are commonly underpowered [88], rendering results of individual studies vulnerable to confounding effects of the chosen study design. In order to circumvent this limitation, we systematically reviewed the available literature on several IEGs in males and females. We meta-analyzed a subset of our male data to quantify cFos expression following ELA exposure and to identify potential moderators of the observed effects. Using a three-level mixed effects model, we observed an increase in cFos expression after an acute stress exposure due to ELA only in combination with one or more other negative life events. This suggests that ELA creates a vulnerable phenotype that is manifested only when sufficiently triggered. If rodents had 'only' experienced ELA, we report–contrary to our expectations–an increase in cFos expression already at rest, suggesting that the situation normally seen (in naïve rodents) after acute stress is already visible at rest when the animals have been exposed to early life adversity. These findings led us to propose a new model as outlined in Fig 6.

At rest, ELA animals compared to controls show increased IEG expression. Since raw values of IEG expression are either not reported or of incomparable scales, we could only investigate effect sizes and not absolute values of IEG expression. This has a direct effect on the interpretation of the results. Specifically, if IEG levels in control animals were low, effect sizes could be inflated. If IEG levels in control animals were high, the results should be interpreted not as "rest" but rather as "mildly aroused", since IEG levels are expected to be minimal for control, naïve animals. Nonetheless, we observed a consistent, positive standardized mean difference in cFos expression after ELA across five out of the six quantitatively investigated brain regions. This suggests a small, but stable brain-wide effect. Previous studies showed that IEG expression matches the transcriptional activity from early environment and experiences [89]. In control animals, this is likely to result in a minimal IEG expression. However, in ELA animals, the expression observed may be the result of long-lasting ELA effects on brain structure and chemistry [90]. Indeed, the transcriptional activity of ELA mice at rest is comparable to that of acutely-stressed control mice [91]. Increased activity-regulated transcription at rest after ELA could be indicative of an overall synaptic alteration, in accordance with increased anxiety-like behavior and reduced memory performance under neutral conditions [16]. Functionally, increased IEG expression at rest could reflect a differential, less adaptive way of processing previous experiences and could potentially hint towards an overall increased transcriptional activity as a result of synaptic sensitization. Intuitively, considering the relationship between IEGs and synaptic plasticity, we would suspect that ELA results in increased synaptic plasticity. In line with this idea, it has been shown that ELA leads to increased LTP in freely-behaving adult, male rats as compared to controls [92].Taken together, this evidence suggests that differences we report in IEG expression after ELA at rest may underlie long-lasting effects on transcriptional activity, pushing the system towards an "activated" state similar to acute stress.

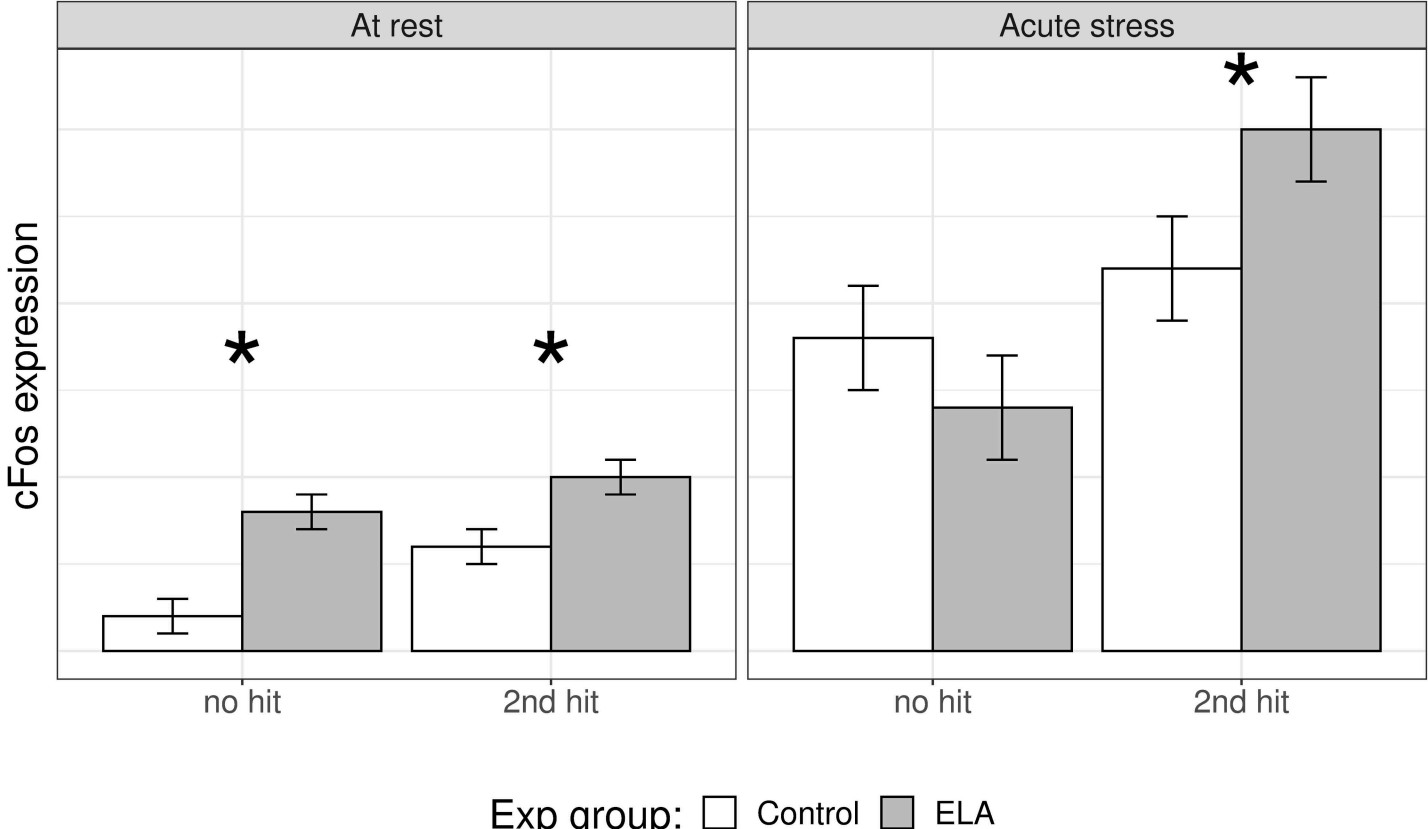

**Fig 6. Summary interpretation of the results.** Cartoon image of how to interpret effect sizes in absolute terms (y-axis, cFos expression, e.g. number of cFos+ cells). Significance levels identify the difference between control and ELA groups that we identified in our analysis (Fig 3). Of note, cFos expression levels are expected to be higher after acute stress than at rest, although this cannot be studied in the current meta-analysis.

The model proposed in Fig 6 relies primarily on the quantitative and qualitative analysis of *cFos* data, as there are only few publications investigating the effects of ELA on the expression of the IEGs *Arc*, *ΔFosB*, and IEGs of the *Egr*-family. Nonetheless, the available evidence suggests a sensitization effect of ELA on IEG expression (and, more generally, synaptic plasticity) at rest. Although IEGs overlap in function and overall expression pattern, they have specific and independent roles [3, 20, 23–25, 93]. *cFos* and *Egr*-family members are transcription factors, while Arc is a post-synaptic protein modifying dendritic AMPA receptors, and *ΔFosB* is a less transient marker of neuronal activity [84, 94, 95]. With technological advances in the field of immunohistochemistry and bioinformatics it becomes increasingly feasible to investigate and interpret multiple IEGs within one animal, thereby also allowing for the investigation of interactions between IEGs and their downstream effects. In the future, the study of different IEGs could be used as a proxy to more thoroughly understand ELA-induced changes in gene-regulated synaptic plasticity [96].

On a systematic review level, effects in females appear more limited than in males. Whether this is a true biological effect is unclear. For example, it could be due to the comparatively low number of female publications, or to a male-biased experimental methodology [16, 76]. While it has been shown that acute stress exposure increases IEG expression in both sexes in the hippocampus [97], it is possible that effects of ELA on IEG expression will be more subtle in females than in males due to model characteristics. Of note, among the 39 publications

included in this review, only 5 investigated males and females under the same experimental conditions.

Lastly, given the expected heterogeneity in study designs, we restricted our meta-analysis to adult animals only, and–at this stage–it cannot be generalized to other age groups. It is possible that different patterns of IEG expression associated to ELA exposure may emerge in juvenile or adolescent animals. Future experiments investigating the longitudinal effects of ELA on IEG expression over the course of development can shed light on the interaction between ELA, development and IEG-related brain activity.

To conclude, we systematically provided a general overview on the relationship between ELA and IEG expression and highlighted current knowledge gaps. Despite subject-specific and methodological limitations, the outcomes of the meta-analysis were robust and suggest a sensitization of activity-regulated transcription in ELA rodents at rest while changes after acute stress only became apparent in combination with additional hits. Recent advances in the fields of immunostaining, live cell imaging and bioinformatics may help close the described voids, yielding a more comprehensive picture on the complex relationship between IEGs, ELA and psychopathologies.

## Supporting information

**S1 File. Supplementary methods, results and references.** S1.1) SYRCLE's study protocol; S1.2) Search string; S1.3) Extracted variables; S1.4) Grouping of variables into functional units; S2.1) Bias assessment; S2.2) Sensitivity analysis species; S2.3) Forest plot; S2.4) systematic review; S3) supplementary references.
(PDF)

**S2 File. PRISMA checklist.**
(PDF)

## Acknowledgments

We would like to thank Eline Kraaijenvanger, Dennis van Nuijs, Lieke van Mourik for their help with articles' selection, and Judith van Luijk for reviewing the study protocol.

## Author Contributions

**Conceptualization:** Heike Schuler, Valeria Bonapersona, Marian Joëls, R. Angela Sarabdjitsingh.

**Data curation:** Heike Schuler, Valeria Bonapersona.

**Formal analysis:** Heike Schuler, Valeria Bonapersona.

**Funding acquisition:** Valeria Bonapersona, Marian Joëls, R. Angela Sarabdjitsingh.

**Investigation:** Heike Schuler, Valeria Bonapersona.

**Methodology:** Valeria Bonapersona.

**Project administration:** Marian Joëls, R. Angela Sarabdjitsingh.

**Resources:** Marian Joëls.

**Software:** Heike Schuler, Valeria Bonapersona.

**Supervision:** Marian Joëls, R. Angela Sarabdjitsingh.

**Visualization:** Heike Schuler, Valeria Bonapersona.

**Writing – original draft:** Heike Schuler, Valeria Bonapersona, Marian Joëls, R. Angela Sarabdjitsingh.

**Writing – review & editing:** Heike Schuler, Valeria Bonapersona, Marian Joëls, R. Angela Sarabdjitsingh.

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
