## [Decision Letter · Decision Letter 0]

6 Aug 2021

PONE-D-21-18961

Effects of early life adversity on immediate early gene expression: systematic review and 3-level meta-analysis of rodent studies

PLOS ONE

Dear Dr. Bonapersona,

Thank you for submitting your manuscript to PLOS ONE. After careful consideration, we feel that it has merit but does not fully meet PLOS ONE’s publication criteria as it currently stands. Therefore, we invite you to submit a revised version of the manuscript that addresses the points raised during the review process.

We look forward to receiving your revised manuscript.

Kind regards,

Alexandra Kavushansky, PhD

Academic Editor

PLOS ONE

“We would like to thank Eline Kraaijenvanger, Dennis van Nuijs, Lieke van Mourik for their help with articles’ selection, and Judith van Luijk for reviewing the study protocol. This work was supported by the Consortium of Individual Development (CID), which is funded through the Gravitation program of the Dutch Ministry of Education, Culture, and Science and the Netherlands Organization for Scientific Research (NWO grant number 024.001.003), and by the ZoNmW program MKMD (grant number 114024135)”

“MJ (Consortium of Individual Development, which is funded through the Gravitation program of the Dutch Ministry of Education, Culture, and Science and the Netherlands Organization for Scientific Research; grant number: 024.001.003).  RAS (ZoNmW program MKMD; grant number 114024135).

Reviewers' comments:

Reviewer's Responses to Questions

**Comments to the Author**

1. Is the manuscript technically sound, and do the data support the conclusions?

Reviewer #1: Partly

Reviewer #2: Partly

Reviewer #3: Yes

2. Has the statistical analysis been performed appropriately and rigorously? 

Reviewer #1: I Don't Know

Reviewer #2: N/A

Reviewer #3: No

3. Have the authors made all data underlying the findings in their manuscript fully available?

Reviewer #1: Yes

Reviewer #2: Yes

Reviewer #3: Yes

4. Is the manuscript presented in an intelligible fashion and written in standard English?

Reviewer #1: Yes

Reviewer #2: Yes

Reviewer #3: Yes

5. Review Comments to the Author

Reviewer #1: Schuler et al. reported effects of early life adversity (ELA) on IEG expression in mice and rats by using a systemic review and meta-analysis. This is an interesting and valuable paper giving a general overview on the relationship between ELA and IEG expression and highlighting current knowledge gaps. There are, however, several issues to be addressed to further improve the manuscript.

1. The authors carefully performed a systemic literature search by listing strict inclusion and exclusion criteria described in the protocol (S1.1). However, no matter how careful and sufficient criteria are set, it would be undeniable that judgment is suffered from subjectivity. Scientific reasons to exclude 2192 publications shown in Fig.2, particularly, wrong outcome, wrong intervention, would be unclear.

2. As the authors mentioned in the Discussion part, the authors restricted their meta-analysis to adult animals only, and they speculated different patterns of IEG expression associated to ELA exposure during different developmental stages. Developing and juvenile stages in rodent brain are critical to neuronal development and HPA-axis programing. Thus, the comparison between these stages and adult stage should be important to understand effects of ELA on IEG.

3. In the supplemental section, the authors mentioned “study inclusion and risk of bias are assessed by two independent researchers, and discrepancies will be resolved by discussion between these two experimenters. Should no conclusion be reached between two experimenters, a third researcher (expert in the field of early life adversity), will be consulted for a solution.” Please clearly show the criteria of expert in the field of early life adversity. Because the role of expert should be critical to the present study.

Reviewer #2: manuscript PONE-D-21-18961

In this study Schuler and Colleagues investigate, by systematic reviewing and meta-analyzing the literature, the long-term effects of early life adversities (ELA) on immediate early genes (IEG) brain expression.

The authors select as ELA the experimental manipulations affecting maternal cares carried out during the three first post-natal weeks and results report IEG expression (Egr family; Arc; �FosB) in different brain areas of mice and rats.

In addition, the summative effects of later single/multiple exposure to stressful conditions in ELA subjects on IEG alteration are also investigated.

Sex-dependent effects and other IEGs than c-Fos (Fos) are not investigated due to small number of publications.

Although this manuscript addresses a really interesting topic and the results could be interesting for the readers of PlosOne, there are some concerns:

Major:

1. Abstract. I think that you have to describe the effects of ELA before the effects of ELA plus acute stress.

2. pg 17, line 282-284. Please rewrite this period, deleting the word “particularly”. Results indicate that cFos expression is increased in ELA animals at rest, while ELA plus acute stressed animals do not show a significant increase.

3. Fig 4. Please add the letters (A,B), clarify in the text the meaning of the words “single”, “multiple” and use the right terms. Please, also use the legend to explain better.

4. The Authors declare in the Method section to restrict the field of investigation focusing on early environmental manipulations sharing the alteration in maternal cares (lines 111-112). Because it is the main factor influencing the literature review/meta-analysis, this should be stressed when appropriate (e.g. title; abstract; introduction).

5 The Authors affirm in Abstract that that they “meta-analyzed publications investigating specifically cFos expression”. Please, focus only on cFos expression (instead of on immediate early gene expression in general) throughout the manuscript. Change the title accordingly.

6. I think that the direction (improve, impairment) of alteration in maternal cares reported in the selected papers should be considered as a critical element.

7. The investigation of stress level (from none to severe) in this study is a precious contribution that Authors could deeply evaluate supporting its role in IEG expression and arranging comparisons throughout all manuscripts.

8. pg 21, line 346-349. I do not agree with the sentence “Increased activity…neutral condition”; you can’t conclude that the effects are related to “sensitization”. Replace the word “sensitization” with a more general term (i.e. alteration) and dampen the conclusions.

Supplementary

9. Please clarify why author choose to evaluate only on a systematic review level “stressors with a strong memory, social or reward component”.

Minor:

Please check the consistency for the terms “single/multiple stress/hit/stressor” throughout all the manuscript and tables.

Line 47-51

Different animal models of perturbation of early life environment fall under the definition of ELA and accordingly a plethora of studies report different long term effects of these depending on the procedure, strain, sex and other variables. I suggest dampening the sentence “ furthermore….and decrease social behavior” (pg 3, line 47-51) because this is only related to one paper from the same group.

Line 69

I suggest to choose between the words “several” and “different”. Use one of the two terms consistently.

Line 113

Please clarify what “grooming and licking” model refer to. Are these selected from natural population variation, selected animal lines and/or genetic driven differences?

Line 214-216

please check the percentage values reported.

Line 334

please check which IEG results (cFos) the sentence refers to.

Add in the table age of testing and time of the tissue collection following stress exposure. It could be useful and informative for readers.

Since stress is a relevant element assessed in present work, please arrange the reported publications in a table according to the categorization of acute stress- from none to mild to severe.

The label “notes” in column is unclear.

Caption

Tab 1 _ I suggest to insert “position” specification at the end of “area acronyms”.

Fig 3 _ Please check the legend (Fem= females ?).

Reviewer #3: Schuler et al have used metanalysis to determine changes in IEGs following ELA.

This is an interesting study which is informative to the field.

Whereas overall the study addresses a common topic is well designed and statistically appropriate, there are several concerns:

1. Early-life adverse experiences are lumped together: The type of early-life stress experienced can have vastly different outcomes on brain development and function, significantly altering behaviour later in life. While the authors describe that they performed sensitivity analysis for type of early-life adversity (ELA) models. This is insufficient, because the majority of studies used maternal separation. it would be interesting to have a better understanding of the ‘other’ group and how these results differed from the rest. If there is no way to properly account for this it would be useful to have some discussion of this in the text.

2. The included studies also varied in the use of stressors. The type of stressor can significantly alter patterns of cfos within the brain (Maras et al 2014), and there is excellent evidence for differential activation of for example, hippocampus by restraint vs footshock. Is it possible to adjust for type of stressor used if not done so already? Or further subdivide this analysis to include only those experiencing a comparable stress?

3. The combining of both mice and rats could be problematic. The authors state that they made the a priori decision to analyse females separately due to known differences. I think the same could be said about mice and rats as they have significant functional and anatomical differences. While much of this would be controlled for by using the within study effect sizes, it could be diluting effects of post-stress activation. Were any analyses for an effect of species performed? Or included as a co-variate?

4. The authors mention some outliers. It would be helpful to inform the reader further about what is different about these studies. Could it be due to any of the above factors, do they use a different model of early life stress?

5. Some of these points could be addressed with the addition of a forest plot detailing the effect sizes for the studies included. Including some annotation or grouping of these differing factors could be informative.

6. PLOS authors have the option to publish the peer review history of their article (what does this mean?). If published, this will include your full peer review and any attached files.

Reviewer #1: No

Reviewer #2: No

Reviewer #3: **Yes: **Tallie Z. Baram

---

## [Author Response · Author response to Decision Letter 0]

20 Sep 2021

New statements for funding and acknowledgements are provided in the cover letter. Point by point answers to the reviewers' comments are available in the "answers to reviewers" file.

---

## [Decision Letter · Decision Letter 1]

23 Nov 2021

PONE-D-21-18961R1Effects of early life adversity on immediate early gene expression: systematic review and 3-level meta-analysis of rodent studiesPLOS ONE

Dear Dr. Bonapersona,

Thank you for submitting your manuscript to PLOS ONE. After careful consideration, we feel that it can be accepted for publication pending minor revisions. Therefore, we invite you to submit a revised version of the manuscript that addresses the points raised by reviewer #3

We look forward to receiving your revised manuscript.

Kind regards,

Patrizia Campolongo

Academic Editor

PLOS ONE

Journal Requirements:

Reviewers' comments:

Reviewer's Responses to Questions

**Comments to the Author**

1. If the authors have adequately addressed your comments raised in a previous round of review and you feel that this manuscript is now acceptable for publication, you may indicate that here to bypass the “Comments to the Author” section, enter your conflict of interest statement in the “Confidential to Editor” section, and submit your "Accept" recommendation.

Reviewer #1: (No Response)

Reviewer #2: All comments have been addressed

Reviewer #3: (No Response)

2. Is the manuscript technically sound, and do the data support the conclusions?

Reviewer #1: Yes

Reviewer #2: Yes

Reviewer #3: Yes

3. Has the statistical analysis been performed appropriately and rigorously? 

Reviewer #1: N/A

Reviewer #2: Yes

Reviewer #3: Yes

4. Have the authors made all data underlying the findings in their manuscript fully available?

Reviewer #1: Yes

Reviewer #2: Yes

Reviewer #3: Yes

5. Is the manuscript presented in an intelligible fashion and written in standard English?

Reviewer #1: Yes

Reviewer #2: Yes

Reviewer #3: (No Response)

6. Review Comments to the Author

Reviewer #1: The author put an effort in revising their manuscript and addressing issues raised previously. Given the nature of the work, several aspects still remain speculative and not convincing. The author’s future analyses should clarify these aspects.

Reviewer #2: The authors have thoughtfully revised this manuscript and addressed all of my concerns. I believe the manuscript is now suitable for publication.

Reviewer #3: The authors addressed most of the concerns in a reasonable manner.

a few additions will be helpful

2. The included studies also varied in the use of stressors. The type of stressor can significantly alter patterns of cfos within the brain (Maras et al 2014), and there is excellent evidence for differential activation of for example, hippocampus by restraint vs footshock. Is it possible to adjust for type of stressor used if not done so already? Or further subdivide this analysis to include only those experiencing a comparable stress?

We thank the reviewer for the critical question. Although different types of acute

stress can cause differential activation throughout the brain, in this meta-analysis we

investigate the difference between controls and ELA animals, rather than the c-fos

distribution within/between brain areas. We reasoned that the effects of ELA may

interact with other factors for acute stressors with a strong memory, social or

reward component. For this reason, we review these experiments only at a

systematic review level (S2.4.3). However, for stressors with a physical component,

we reasoned that the difference in effects between ELA and control should be

comparable. In the 28 experiments using acute stressors, 10 different “types” of

acute stressors were used. Due to the heterogeneity, it is not possible to quantify

the different types with a subgroup analysis, although we did further categorize

them (S1.4B) into mild/severe for subgroup analysis (results reported in lines 287-

290). Additionally, the column ‘Exp design details’ in Table 1 specifies the stressor

type of each experiment, such that interested reader can qualitatively assess the

effects of a stressor of interest.

It would be useful to include the cross-reference to the definition in the supplement of mild/severe in the relevant results section. I couldn’t find the relevant report on lines 287-290 and assume this is an error and the authors are referring to the analysis on lines 308-312 and think this information should be included here. The authors reasoning regarding the choice of stressors included in the analyses (as detailed above) and the heterogeneity should be included in the correct section.

4. The authors mention some outliers. It would be helpful to inform the reader further

about what is different about these studies. Could it be due to any of the above factors,

do they use a different model of early life stress?

One comparison was excluded as an influential outlier, following a conservative

statistical approach as described in reference [36]. The study used maternal

separation in male rats and the comparison looked at cFos expression in the

hypothalamus after restraint stress; the publication was not from a predatory

journal. No element of the experimental design “stood out”, and we therefore

interpreted this particular comparison as a statistical (rather than biological) outlier.

We have added the following in line 178 to clarify the outlier removal procedure:

“[Influential outliers were determined in accordance with Viechtbauer and Cheung

[36] and removed from quantitative synthesis. Of such comparisons, we explored

whether elements of the experimental design could explain the deviation of these

comparisons from the mean.”

In addition, we included qualitative information on the outlier in line 232-234:

“No element of the experimental design pointed towards a biological origin of the

outlying value, nor was its publication published in a predatory journal.”

Thanks for the added information, it may be useful to include information regarding the definition of predatory journal that was used eg. https://www.nature.com/articles/d41586-019-03759-y

7. PLOS authors have the option to publish the peer review history of their article (what does this mean?). If published, this will include your full peer review and any attached files.

Reviewer #1: **Yes: **Mayumi Nishi

Reviewer #2: No

Reviewer #3: No

---

## [Author Response · Author response to Decision Letter 1]

10 Dec 2021

Response to Editor comments: 

All references have been doubled checked, and we do not reference any article that has been retracted.

Response to Reviewer's comments: 

Please see rebuttal letter attached to the submission.

---

## [Editor Report · Decision Letter 2]

13 Dec 2021

Effects of early life adversity on immediate early gene expression: systematic review and 3-level meta-analysis of rodent studies

PONE-D-21-18961R2

Dear Dr. Bonapersona,

We’re pleased to inform you that your manuscript has been judged scientifically suitable for publication and will be formally accepted for publication once it meets all outstanding technical requirements.

Kind regards,

Patrizia Campolongo

Academic Editor

PLOS ONE
---

## [Editor Report · Acceptance letter]

3 Jan 2022

PONE-D-21-18961R2 

Effects of early life adversity on immediate early gene expression: systematic review and 3-level meta-analysis of rodent studies 

Dear Dr. Bonapersona:

I'm pleased to inform you that your manuscript has been deemed suitable for publication in PLOS ONE. Congratulations! Your manuscript is now with our production department. 

Kind regards, 

on behalf of

Dr. Patrizia Campolongo 

Academic Editor

PLOS ONE